# Exploring the Quality of Life of People with Incomplete Spinal Cord Injury Who Can Ambulate

Murveena Jeawon [1,2,3], Bethany Hase [1], Susanna Miller [1], Janice Eng [2,3,4,5], Andrea Bundon [3,6], Habib Chaudhury [7], Jocelyn Maffin [8], Ryan Clarkson [8], Jenna Wright [8] and W. Ben Mortenson [1,2,3,5,*]

1  Department of Occupational Sciences and Occupational Therapy, University of British Columbia, Vancouver, BC V6T 2B5, Canada; murveena.jeawon@ubc.ca (M.J.); bethany.hase@alumni.ubc.ca (B.H.); susanna.miller@vch.ca (S.M.)
2  GF Strong Rehabilitation Centre, Vancouver, BC V5Z 2G9, Canada; janice.eng@ubc.ca
3  International Collaboration on Repair Discoveries, Vancouver, BC V5Z 1M9, Canada; andrea.bundon@ubc.ca
4  Department of Physical Therapy, University of British Columbia, Vancouver, BC V6T 1Z3, Canada
5  Centre for Aging SMART, Vancouver Coastal Health, Vancouver, BC V5Z 1M9, Canada
6  School of Kinesiology, University of British Columbia, Vancouver, BC V6T 1Z1, Canada
7  Department of Gerontology, Simon Fraser University, Vancouver, BC V6B 5K3, Canada; chaudhury@sfu.ca
8  Spinal Cord Injury—British Columbia, Vancouver, BC V6P 5Y7, Canada; jmaffin@sci-bc.ca (J.M.); rclarkson@sci-bc.ca (R.C.); jennawri.jw@gmail.com (J.W.)
*  Correspondence: ben.mortenson@ubc.ca

**Abstract:** (1) Purpose: To examine associations between subjective quality of life and other socio-demographic variables and to explore differences in experiences of people with different levels of quality of life (low, moderate, high). (2) Materials and methods: Semi-structured interviews and standardized measures of mobility, function, health-related quality-of-life, and quality-of-life were used to collect the data for this mixed-method study. (3) Results: Twenty-four participants were interviewed with an average age of 55 years and 54% were male. High quality of life, according to quantitative analysis, was strongly associated with being male, attending rehabilitation, and being married. The qualitative findings supported the quantitative findings and also revealed that people with a low quality of life felt the neighborhood-built environment was not supportive of people with incomplete spinal cord injury who can walk. Participants who reported a low/moderate quality of life reported feeling devalued by able-bodied people and that their mobility was getting worse over time. (4) Conclusion: Findings suggest that those with incomplete spinal cord injuries who can walk could benefit from improved quality of life by modifying their social support and neighborhood's built environment. For instance, sensitivity training for the general population could help to reduce negative attitudes and misperceptions about invisible impairments and promote inclusion.

**Keywords:** quality of life; experiences; invisible impairments; social scrutiny; excluded; resilience

## 1. Introduction

Spinal cord injury is relatively common and can have life-altering effects. In Canada, there are approximately 86,000 people with spinal cord injuries, with 4300 new cases each year [1,2]. The number of Canadians with incomplete spinal cord injuries is increasing [3]. As Canada's population ages, in 10 years the number of people living with spinal cord injury will likely rise to more than 120,000 [2]. In Canada, 52% of people have incomplete tetraplegia, 18% have incomplete paraplegia and the remainder (30%) have complete paraplegia or tetraplegia [4]. Depending on the extent of their injury, people with incomplete spinal cord injury frequently regain some form of functional ambulation thanks to improvements in acute rehabilitation [1,5,6]. For example, people with an American Spinal Cord Injury Association Impairment Scale score of C or D are likely to ambulate [7].

Empirical evidence suggests that people with an incomplete spinal cord injury who can ambulate struggle with physical and psychological issues differently from those with

complete injuries. Although many people want to walk after spinal cord injury, a cross-sectional cohort study found that long-term ambulation post spinal cord injury was related to undesirable outcomes such as fatigue, pain, and depressive symptoms in 56% of the study participants [8]. Several studies have suggested that the relationship between the ability to walk and life satisfaction may be mediated by both physical factors (pain, fatigue) and psychological factors (depression, frustration) [9–11]. A survey study identified having an incomplete injury and poor mental well-being as main factors associated with depression [12]. Another survey found that people with an American Spinal Cord Injury Association Impairment Scale score of D had lower social participation compared to those with scores of A, B, or C [13].

Various demographic factors have been associated with a positive quality of life (defined as how people perceive their emotional and physical well-being which encompasses a wide range of areas, including corporal, emotional, economical, devotional, and social prosperity [14]) in combined samples (i.e., those with complete and incomplete injuries). Three systematic reviews [15–17], a cross-sectional study [18], a case-control study [19], and an empirical research study [20], identified a range of factors that positively influenced life satisfaction, including getting injured at a younger age, having a lower level of spinal cord injury, being employed, being married, not living in poverty, living in urban areas, getting more leisure-time physical activity, keeping a positive attitude, enjoying accessible transportation, time since injury and having recovered from any previous health issues.

Our review of the literature identified few studies exploring the quality of life among people with spinal cord injury who can ambulate. A cross-sectional survey study conducted in Norway suggested that people with incomplete spinal cord injuries who participated in physical activities (e.g., walking, biking, strength training) had higher life satisfaction [21]. People with an American Spinal Cord Injury Association Impairment Scale of D were substantially less happy with their mental health compared to people with an American Spinal Cord Injury Association Impairment Scale of A/B/C as reported by a Sweden cross-sectional cohort study [22]. A cross-sectional study from the United States reported that veterans with an American Spinal Cord Injury Association Impairment Scale score of D reported more pain, depressive symptoms, and general poor health compared to those with other American Spinal Cord Injury Association Impairment Scale scores [23]. A qualitative study conducted in Australia suggested this difference may be due to a variety of potential causes such as being annoyed with the speed of completing daily activities, being exhausted from walking, feeling misunderstood owing to invisible impairment leading to less support from the community, and having symptoms overlooked [11].

Given the limited understanding of the quality of life of people with incomplete spinal cord injury who can ambulate, especially from a Canadian perspective, we conducted this study with two main objectives: (1) to examine associations between subjective quality of life and other socio-demographic variables in this population and (2) to explore differences in experiences of people with different levels of quality of life (low, moderate, and high) using a novel mixed-methods approach.

## 2. Methods

This study used an explanatory design in which qualitative data were used to explain the quantitative findings; i.e., the qualitative findings were built upon the quantitative findings [24,25]. Specifically, we used the quantitative data to identify participants with three levels of quality of life (low, moderate, and high) and to compare sociodemographic characteristics in those three groups. We explored the experiences of participants in each of these groups qualitatively.

Qualitative description was the method used to analyze the qualitative data set and the findings were reported using the consolidated criteria for reporting qualitative research guidelines [26,27] (Table S1). The overall study has been reported using the Good Reporting of a Mixed Methods Study Framework [28] as reported in (Table S2).

*2.1. Study Participants*

Participants were eligible if they were: (1) diagnosed with incomplete spinal cord injury, (2) able to walk 15 m with or without aids short distances, (3) reside in British Columbia, (4) 19 years or older, (5) able to understand and to read English and (6) able to provide their own consent.

*2.2. Recruitment*

We recruited a convenience sample of 24 participants and there were no dropouts. The study was advertised on the International Collaboration on Repair Discoveries website, on the Mortenson Lab Social media channels, at the GF Strong Rehabilitation Centre, and in the Spin Magazine that was distributed by Spinal Cord Injury—British Columbia quarterly. Previous study participants who indicated an interest in being notified about future studies were also contacted.

*2.3. Data Collection*

Participants were recruited over a period of five months and provided informed written consent to participate in the study. Due to the COVID-19 pandemic, data were collected virtually (i.e., via Zoom [29] or telephone). Before the interview, participants completed the standardized measures and demographic survey on an online survey program (Qualtrics [30]). All audio was digitally recorded on the interviewer's computer using a University of British Columbia Zoom account and was transcribed verbatim. All transcripts were password-protected. Prior to the interviews, participants were emailed detailed information about how to install and use Zoom (a direct link to download was provided). In case of any difficulties during the Zoom meeting (e.g., loss of internet connection) participants were contacted on their phones to continue with the interview. Participants were reminded to be alone in the room during the interview. If the participant was unable to use Zoom because of lack of internet and or difficulty with hand function, the three student authors administered the interview orally over the phone. The study was an unblinded qualitative study.

*2.4. Measures*

Participants completed one demographic survey and four standardized measures as described below.

2.4.1. Demographic Survey

A demographic form was used to gather descriptive information. This survey collected data about participants' age, sex at birth, type of spinal cord injury, number of years living with injury, employment status, living situation, education level, and marital status.

2.4.2. Functional Ambulation Measure

Participants reported their ambulatory ability according to the Functional Ambulation Categories [31]. The six levels are; non-functional ambulator (level 0); ambulatory dependent on physical assistance (level 1—indicates a patient who requires continuous manual contact to support body weight as well as to maintain balance or to assist coordination); ambulator dependent on physical assistance (level 2—Indicates a patient who requires intermittent or continuous light touch to assist balance or coordination); ambulator dependent on supervision (level 3); ambulator independent on flat surface only (level 4) and independent ambulator (level 5). A score of 0 indicates one cannot walk and a score of 5 indicates one can walk freely.

2.4.3. Life Satisfaction Questionnaire—11

Quality of life was assessed using the life satisfaction questionnaire [32] consisting of 11 questions on specific domains. The 11 items in the questionnaire are life as a whole, work, finance, leisure, contact with friends, sexual life, daily activities, family life, partner

relationships, somatic health, and psychological health. Each item gets a score of 1 to 6 depending on how satisfied the person feels. A score of 1 is very dissatisfied and a score of 6 is very satisfied [33].

### 2.4.4. Spinal Cord Independence Measure—III

Function was assessed using the self-report version of the Spinal cord independence measure [34] of 19 items divided into three categories: respiration and sphincter management, self-care, and mobility. Each item obtains a score of 0–100, where a score of 0 indicates total dependence and 100 indicates complete independence.

### 2.4.5. 12-Item Short-Form Survey

Health-related quality of life was measured using the 12-item short-form health survey [35] consisting of 12 questions covering eight specific domains. This is a shorter version of the 36-item short-form health survey. The eight domains are mental health, physical role, bodily pain, social functioning, vitality, general health perspective, physical functioning, and emotional health. Each item obtains a score of 0–100, with 0 indicating poor health and 100 indicating very good health. There are two summary scores used to calculate the overall score of the 12-item short-form health survey namely the physical component score and the mental component score.

### 2.4.6. Semi-Structured Interviews

Semi-structured interviews were conducted after participants had completed the online survey. Interviews were conducted only once by three graduate students trained in qualitative research (The lead author who was a female Master's student with training in occupational therapy and two co-authors, who were female Occupational Therapy Master's students). They were guided by the student supervisor and senior authors who have extensive experience with mixed-methods research. Interviews lasted between 45 to 90 min and were conducted one-on-one (one interview and one interviewee). Participants were instructed to be alone in the room during the interview. No relationship was developed prior to the interviews with the participants and none of them were familiar with any of the interviewers. The interview guide (Document S1) was developed and piloted collaboratively with the research team and three co-authors who were working for a non-profit organization that provides assistance to people with spinal cord injuries.

### 2.5. Data Analysis

### 2.5.1. Quantitative

To address the first research question, the lead author divided participants into three levels of quality of life with the following cut points: low ($\leq$28), moderate (29–47), and high ($\geq$48). These cut points were used so that we had a relatively equal number of people in each group. Descriptive statistics data including mean, standard deviation, number, and percentage for categorical and continuous variables were used to characterize participants in each group.

To test for the normality of data, we conducted Kolmogorov–Smirnov tests. Three scores were abnormally distributed, the physical component score of the 12-item short-form health survey ($p = 0.05$), the spinal cord independence measure ($p = 0.02$), and the functional ambulatory measure ($p \leq 0.01$).

We looked at the relationship between quality of life and other sociodemographic variables on a bivariate level rather than comparing the three levels of quality of life. A Spearman correlation was used for continuous variables and Mann–Whitney U tests or Kruskal–Wallis tests were used for categorical data. A significance level of $p \leq 0.05$ was used for all tests; however, it is important to consider the strength of associations [36]. In terms of effect sizes for Spearman's rho, a negligible relationship is $r_s = 0.00–0.20$; a weak relationship is $r_s = 0.21–0.40$; a moderate relationship is $r_s = 0.41–0.60$; a strong relationship is $r_s = 0.61–0.80$ and a very strong relationship is $r_s = 0.81–1.00$ [37]. For Cohen's d a small

effect size is d = 0.20, a medium-size effect is d = 0.50 and a large-size effect is d = 0.80 [38]. For Cohen's f, a small effect is f = 0.10, a medium effect is f = 0.25 and a large effect is f = 0.40 [36]. To understand better the relationship between sex, marital status, and living situation we plotted the interactions between sex and living situation, sex and marital status, and living situation and marital status (Figure A1). We were not able to produce a plot for the interaction between living situation and marital status because the sample size was too small. Data were analyzed using the IBM SPSS software [39].

### 2.5.2. Qualitative

To analyze the data, the transcripts for each participant were reviewed by the lead author to identify potential barriers and facilitators to their quality of life. The data were managed in NVivo [40], and then a table documenting potential barriers and facilitators to quality of life in each group (i.e., low, moderate, and high) was made.

To promote the rigor of the study we used the four transcendent trustworthiness criteria identified by Morrow [37]: (1) social validity, (2) reflexivity and subjectivity, (3) adequacy of data, and (4) adequacy of interpretation. In terms of social validity, the study was suggested by people working at a non-profit organization for people with spinal cord injuries. They were part of the team as co-authors and have lived experience of spinal cord injury. Reflexivity and subjectivity require researchers to consider their positioning when collecting and interpreting data [41]. The three student authors conducting the interviews maintained reflexive journals that documented their reflections and personal positioning. For example, they recorded how they anticipated participants would respond to each interview question. They probed for negative cases if participants were telling them things they expected to hear.

Adequacy of data is concerned with the quantity and quality of data, as having a greater number of participants would guarantee the validity of the findings. Some argue that the collection of data should be conducted until the point of redundancy when there is no more new information to be acquired from the ongoing data collection. Redundancy also known as saturation happens when no new information emerges but questions have been raised about how this idea is operationalized [42]. Others have recommended sufficiency rather than saturation with qualitative research [43–45].

Adequacy of interpretation is to involve multiple researchers in the data collection and analysis so that their complementary perspectives would enrich the findings. To promote adequacy of interpretation we addressed these criteria by doing the following: involvement of multiple researchers in data collection and analysis; creation of a casual and trusting relationship to try and make participants feel comfortable (by being friendly as opposed to being just an interviewer, and when the interview finished, we let the participant continue talking about anything they wanted to add/share). Previous research has suggested that some participants are more relaxed and confident to talk when they are in their safe space (in their house) [46]. We shared the summary of the results with the participants and we requested feedback to which five of them responded and all agreed to our key conclusion.

## 3. Results

### 3.1. Quantitative

Considering the sample as a whole (n = 24), it was noted in Table 1 that there were slightly more male than female (row 2; column 3). On average, most of the participants had a traumatic spinal cord injury (row 4; column 3) and most of them attended rehabilitation (row 9; column 3) after injury. Only four participants were employed after injury (row 11; column 3) and a majority lived with somebody (row 19; column 3). According to the three different levels of quality of life (low n = 5, moderate n = 9, high n = 10), the majority of participants in the low quality of life group were females (row 3; column 4) whereas the majority of participants in the high quality of life group were males (row 2; column 6).

**Table 1.** Mann–Whitney U, Kruskal–Wallis, Spearman correlation, Cohen's d, Cohen's f with demographic factors, standardized measures, and 3 levels of quality of life.

| Variable | Label (Potential Range) | [Mean ± SD]/n (%) | Low Quality of Life (n = 5) n (%)/Mean ± SD | Moderate Quality of Life (n = 9) n (%)/ [Mean ± SD] | High Quality of Life (n = 10) n (%)/ [Mean ± SD] | Statistic [1] U/[2] (H)/[3] [$r_s$] | Cohen's d/(f) | p |
|---|---|---|---|---|---|---|---|---|
| Sex | Male | 13 (54.16) [48.31 ± 10.68] | 1 (20.00) | 4 (44.44) | 8 (80.00) | 30.00 [1] | 1.12 | 0.02 |
| | Female | 11 (45.83) [34.82 ± 13.28] | 4 (80.00) | 5 (55.56) | 2 (20.00) | | | |
| Type of injury [4] | Traumatic | 14 (58.30) [45.79 ± 13.05] | 2 (40.00) | 4(44.44) | 8 (80.00) | 32.00 [1] | 0.29 | 0.41 |
| | Non Traumatic | 6 (25.00) [40.67± 14.62] | 1 (20.00) | 3(33.33) | 2 (20.00) | | | |
| Level of spinal cord injury | Cervical | 9 (37.50) | - | 4 (44.44) | 4 (40.00) | (0.58) [2] | (0.03) | 0.75 |
| | Thoracic | 6 (25.00) | 2 (40.00) | 2 (22.22) | 3 (30.00) | | | |
| | Lumbar | 9 (37.50) | 3 (60.00) | 3 (33.33) | 3 (30.00) | | | |
| Attended rehabilitation | Yes | 16 (66.67) [46.38 ± 10.36] | 1 (20.00) | 7 (77.78) | 8 (80.00) | 29.00 [1] | 0.957 | 0.032 |
| | No | 8 (33.33) [33.62 ± 15.76] | 4 (80.00) | 2 (22.22) | 2 (20.00) | | | |
| Employed | Yes | 4 (16.67) [43.75 ± 16.86] | 1 (20.00) | 1 (11.11) | 2 (20.00) | 34.50 [1] | 0.13 | 0.67 |
| | No | 20 (83.33) [41.80 ± 13.28] | 4 (80.00) | 8 (88.89) | 8(80.00) | | | |
| Marital Status | Never married/separated/ Divorced | 11 (45.83) [35.27 ± 14.16] | 5 (100.00) | 3 (33.33) | 3 (30.00) | 37.00 [1] | 1.02 | 0.04 |
| | Married/Common Law | 13 (54.17) [47.92 ± 10.24] | - | 6 (66.67) | 7 (70.00) | | | |
| Location of residence | City | 13 (54.16) | 3 (60.00) | 3 (33.33) | 7 (70.00) | (1.49) [2] | (0.06) | 0.48 |
| | Suburban | 7 (29.17) | - | 5 (55.56) | 2 (20.00) | | | |
| | Rural | 4 (16.67) | 2 (40.00) | 1 (11.11) | 1 (10.00) | | | |
| Living situation | Live alone | 8 (33.33) [37.00 ± 14.21] | 3 (60.00) | 3 (33.33) | 2 (20.00) | 44.50 [1] | 0.57 | 0.23 |
| | Live with somebody | 16 (66.67) [44.69 ± 12.88] | 2 (40.00) | 6 (66.67) | 8 (80.00) | | | |
| Level of education | 2 = High school | 7 (29.17) | [3.60 ± 1.14] | [3.56 ± 1.51] | [3.70 ± 1.57] | [0.07] [3] | - | 0.72 |
| | 3 = College/Trade School | 5 (20.83) | | | | | | |
| | 4 = University degree | 5 (20.83) | | | | | | |
| | 5 = Graduate studies | 4 (16.67) | | | | | | |
| | 6 = Postgraduate | 3 (12.50) | | | | | | |
| Number of Years Living with an injury | (2–52) | [21.88 ± 16.30] | [26.20 ± 16.92] | [27.67 ± 18.53] | [14.50 ± 12.01] | [−0.40] [3] | - | 0.05 |
| Gross annual income [5] | (1–7) [6] | [1.83 ± 2.88] | [2.20 ± 2.39] | [0.78 ± 2.39] | [2.60 ± 3.44] | [0.15] [3] | - | 0.48 |
| Age (Years) | (27–72) | [54.60 ± 14.60] | [50.60 ± 15.44] | [54.22 ± 12.17] | [57.30 ± 16.95] | [0.19] [3] | - | 0.37 |

**Table 1.** *Cont.*

| Variable | Label (Potential Range) | [Mean ± SD]/n (%) | Low Quality of Life (n = 5) n (%)/Mean ± SD | Moderate Quality of Life (n = 9) n (%)/ [Mean ± SD] | High Quality of Life (n = 10) n (%)/ [Mean ± SD] | Statistic [1] U/[2] (H)/[3] [r_s] | Cohen's d/(f) | p |
|---|---|---|---|---|---|---|---|---|
| 12-Item Short-form health survey: Mental component scores | (22–60) | [39.95 ± 11.11] | [29.77 ± 5.29] | [36.04 ± 7.34] | [48.56 ± 9.99] | [0.74] [3] | - | <0.01 |
| 12-Item Short-form health survey:Physical component scores | (20–57) | [35.10 ± 9.86] | [28.99 ± 10.36] | [29.95 ± 4.63] | [42.80 ± 8.27] | [0.68] [3] | - | <0.01 |
| Spinal cord independence measure—III | (38–100) | [79.04 ± 13.90] | [70.20 ± 18.19] | [75.44 ± 13.72] | [86.70 ± 7.62] | [0.34] [3] | - | 0.11 |
| Functional ambulatory category | 4 = Ambulate independently level surface only | 12 (50.00) [38.50 ± 12.52] | 3(60.00) | 6(66.67) | 3(30.00) | 46.50 [1] | 0.54 | 0.14 |
| | 5 = Ambulate independently | 12 (50.00) [45.75 ± 14.08] | 2(40.00) | 3(33.33) | 7(70.00) | | | |
| Life satisfaction questionnaire—11 | (20–64) | [42.13 ± 13.54] | [22 ± 2.35] | [39 ± 4.44] | [55 ± 5.46] | - | - | - |

[1] U: Mann–Whitney U. [2] H: Kruskal–Wallis. [3] r_s: Spearman correlation. [4] Type of injury: four participants did not know the type of injury. [5] Gross income: 10 participants preferred not to answer. [6] 1 = <$14,999; 2 = $15,000–$29,999; 3 = $30,000–$44,999; 4 = $45,000–$59,999; 5 = $60,000–$74,999; 6 = >$75,000; 7 = Prefer not to answer.

The two columns (column 7 and column 8) reported the correlation and effect size. A strong correlation was found between the life satisfaction questionnaire and the 12-item short-form health survey (physical component scores and mental component scores). Males, people who attended rehabilitation, and people who were married appeared to have a higher quality of life (large effect sizes).

Not shown in Table 1, based on (Figure A1) there appeared to be an interaction between sex and living situation. Males and females who lived alone had similar low quality of life. Males living with someone else had the highest quality of life, whereas females living with others had the lowest quality of life. Comparing sex and marital status, males who were married had the highest quality of life whereas females who were married seemed to have similar quality of life as single males. Unmarried females had a lower quality of life. Due to a small sample size, we were unable to generate a plot to explore the relationship between marital status and living situation.

*3.2. Qualitative*

Table 2, describes qualitative barriers and facilitators to quality of life by reviewing participants' lived experiences. People in the low quality of life group experienced more barriers compared to those in the high quality of life group who had more facilitators and very few barriers. A balance between facilitators and barriers to quality of life was identified in the moderate quality of life group. Some issues were facilitators in one group and barriers in another group, e.g., social support is a facilitator in the high-quality-of-life group but is a barrier in the low-quality-of-life group. Other issues seemed to overlap many times in between groups, e.g., learning to adapt to life after the injury is a facilitator for

both groups in moderate and high quality of life, and feeling judged is a barrier for both groups in low and moderate quality of life.

**Table 2.** Potential reason for the quality of life ratings.

| | Low Quality of Life (P1, P6, P11, P13, P14) | Moderate Quality of Life (P4, P5, P7, P9, P10, P15, P16, P18, P20) | High Quality of Life (P3, P2, P8, P17, P12, P19, P21, P22, P23, P24) |
|---|---|---|---|
| Potential **Barriers** to Quality of Life | • Feeling socially isolated<br>• Having financial concerns<br>• Feeling judged by abled-bodied people<br>• Feeling left out due to City layout and buildings not being wheelchair friendly<br>• Expressing that spinal cord injury was due to medical negligence<br>• Describing their deteriorating mobility<br>• Feeling left out due to lack of resources for people with disabilities | • Feeling judged<br>• Describing their deteriorating mobility<br>• Feeling socially isolated<br>  ○ COVID-19 restriction<br>• Having financial concerns<br>• Feeling left out due to:<br>  ○ Lack of resources for people with disabilities<br>  ○ City layout and buildings not being wheelchair friendly<br>  ○ Spinal cord injury programs not designed for people with complete spinal cord injury<br>• Describing secondary complications of spinal cord injury<br>• Passing as a person with disabilities | • Feeling left out due to:<br>  ○ City layout and buildings not being wheelchair friendly<br>  ○ Spinal cord injury programs not designed for people with complete spinal cord injury spinal<br>• Feeling frustrated by COVID-19 restrictions |
| Potential **Facilitators** to Quality of Life | • Learning to adapt to life after injury | • Learning to adapt to life after injury<br>• Appreciating their support system:<br>  ○ Having social support<br>• Having financial stability<br>• Feeling included in spinal cord injury programs<br>• Benefiting from COVID-19 restrictions such as<br>  ○ Improved financial situation<br>  ○ Improved access to medical carel<br>• Benefiting from:<br>  ○ Adapted/modified house<br>  ○ Wheelchair-friendly city layout and buildings<br>• Expressing gratitude to still be able to walk | • Learning to adapt to life after injury:<br>  ○ Demonstrating resilience<br>• Appreciating their support system:<br>  ○ Feeling included in spinal cord injury programs and community<br>  ○ Having family support<br>  ○ Having Social support<br>• Experiencing little difference with day to day life during COVID-19 restrictions<br>• Having financial stability<br>• Benefiting from<br>  ○ Adapted/modified house<br>  ○ Wheelchair-friendly city layout and buildings<br>• Benefiting from COVID-19 restrictions such as<br>  ○ Improved social life<br>  ○ Improved access to medical care<br>• Being able to ambulate without an assistive device<br>• Describing a sense of gratitude |

### 3.2.1. Low Quality of Life (n = 5)

Participants in this group described six factors that they perceived negatively affected their quality of life: feeling socially isolated, having financial concerns, feeling judged, feeling left out due to city layout and infrastructure not being wheelchair or walker-friendly, expressing that spinal cord injury was caused due to medical negligence and describing

their deteriorating mobility. In this group, only one participant described one factor (ability to adapt to injury) that positively impacted their quality of life.

Three participants described how their friends ended their friendship following their injuries. As noted in Table 3, P11 attributed this to problems she had with pain and not obtaining equipment in a timely manner. With the start of the COVID-19 restrictions, all the participants in the low-quality-of-life group felt extremely isolated for a variety of reasons. For example, three participants lived alone or were restricted to limited social interactions due to travel restrictions, quarantine measures, and the fear of contracting the disease. Despite the possibility of virtual communication, these participants preferred in-person interactions as expressed by participant P14 in Table 3. Participant P06, a 31-year-old male injured in the lumbar region described how the pandemic restrictions made him feel more socially isolated. He had a small group of friends and they had stopped seeing each other because of COVID-19 restrictions. As the pandemic progressed his group of friends grew smaller, which made him spend more time playing games on his computer.

Four participants indicated that their limited income led to unmet medical needs, and the inability to acquire important assistive devices such as a foot orthosis or a powered wheelchair. Most of them were unemployed (n = 4) and depended only on their disability pension, so they explained how expensive medical equipment was, as noted by Participant P06 in Table 3.

All participants in the low quality of life expressed concerns about very expensive medical equipment. Participant P14 suggested removing taxes on medical equipment in order to increase the affordability of equipment that people need in terms of daily living to be able to have a better quality of life.

Participants reported three types of social-environmental barriers to quality of life as a result of being judged by abled-bodied people: being invisible, being hyper-visible, and being treated inhumanely. All participants in the low-quality-of-life group reported feeling ignored often by able-bodied people. P14 expressed how they made her feel invisible (Table 3). All five participants described being stigmatized which they attributed to the use of their mobility device or having an impairment that was visible to others (e.g., limping). P14 reported how she experienced prejudice (Table 3). Two participants related some events during which they felt they were being treated inhumanely. Participant 13 a female, 68 years old, who was injured at L3–L5, had to stop going to pottery workshops because she felt excluded by others who attended. Participant P06 described that other people perceived him as a "half-person" because he uses a wheelchair which could cause some people to look down on him figuratively and literally.

More than half of the participants in this group experienced the loss of independence because of the city's unsupportive infrastructure such as having no curb cuts, narrow sidewalks, the presence of cobblestone/stairs, hilly topography, and broken elevators. Participant P11 reported how accessibility challenges made it difficult to get around (Table 3). Participants explained that some places that were supposedly accessible turned out to be inaccessible. Participant P14 indicated that a lot of buildings are not completely accessible for people with a disability, because they do not have an accessible washroom or have multi-levels with no functioning elevators. Four out of five participants indicated they did not have any formal rehabilitation after their injury, which they felt could have improved their lives considerably. Participant P01 tried multiple times to access rehabilitation and was very vocal that if she had not been denied care, her health would not have deteriorated (Table 3).

All participants in this group indicated how ongoing secondary health conditions (e.g., ongoing pain, spasticity, and tiredness) made daily life challenging (e.g., simple activities required a lot of planning, extra energy, and necessitated rest) and caused degradation in mobility. Participant P13 reported that she felt like she was prematurely aging (Table 3). Three participants reported how their ability to walk fluctuated and ultimately deteriorated after their injury. Participant P06 indicated his gait had worsened over time post-injury. He

started walking with a cane, progressed to using no cane, then regressed to a walker, and now uses a wheelchair for outdoor activities instead of his walker.

**Table 3.** Low life satisfaction quotes organized by factor.

| Topic | Quotation | Participant Information |
|---|---|---|
| **Feeling socially isolated** | | |
| Losing friends because of pain intolerance | 'I have ongoing and significant pain, difficulty sitting, and difficulty walking. I can't sit in an upright position for very long without pain, I can sit in my wheelchair for about two hours but I couldn't sit in a regular kitchen chair on a couch. I wasn't able to go to see friends anymore for lunch or coffee because I couldn't sit. I went a long time without a wheelchair before I was finally approved for one.' | P11, female, 63 years old, T6 |
| Losing human contact and preferred in-person interaction | 'I miss some of the important people in my life because we are not able to connect in the same way. Phone calls are not the same as actually going out and meeting people, person to person, and doing something fun together or going out for a meal or hanging out at a social event, it's not the same connection.' | P14, female, 51 years old, thoracic |
| **Having financial concerns** | | |
| Identifying financial need as a barrier to participation | 'That's been a barrier because of the low income that I get from my pension. If I wanted to get an electric assistive device this is kind of expensive, I do need home adaptations right now, but those cost a high price so I have to adapt myself to my house set up for the time being which is tiring and consumes most of my energy.' | P06, male, 31-year-old, injured in the lumbar region |
| **Being judged** | | |
| Feeling invisible | 'That's pretty much an everyday occurrence. People treat me like this because I'm physically disabled and think I'm mentally disabled and they'll ask the person I'm with the question, instead of asking me, which is very infuriating because I'm an adult and you know, highly competent, I can talk for myself.' | P14, female, 51 years old, thoracic |
| Experiencing prejudice | 'Discrimination happens all the time because people have attitudinal barriers. They're straight-up rude and basically kind of treat you like you're a non-human. And that pretty much is all the time because I look physically disabled when I'm out and about, I get stares, I get looks, it's pretty much constant.' | P14, female, 51 years old, thoracic |
| **Feeling left out due to city layout and infrastructure** | | |
| Experiencing accessibility challenges in the built environment | 'The area where I live is a bit hilly. So it's difficult either using my walker/my cane/my wheelchair; there are some streets I can't go on. I find that some of those curb cuts on the sidewalk are steep and sometimes I have to get out of my wheelchair and push it up the curb cuts, then I can get back in. Some of the stores in the area I live have a couple of steps up or down, to get into the stores, so of course, I don't go there. If a place is not wheelchair accessible I don't go there.' | P11, female, 63 years old, T6 |
| **Expressing that spinal cord injury was due to medical negligence** | | |
| Expressing deterioration in health due to medical negligence | 'If I would've had the machine to help me breathe, my health, my walking, and everything else would never have deteriorated. I never had to end up in the hospital and now I have to go to the hospital to see a cardiologist, All because I didn't get the right rehabilitation for my deteriorating leg, hip, and back muscles with the machine to help me breathe.' | P01, female, 40 years old, T6–T7 |
| **Expressing a degradation in mobility** | | |
| Feeling like prematurely aging | 'It's like living with a 121-year-old woman, that's what I'm like. I bring in the groceries from the car using the walker, I'll be paralyzed in bed for 8–14 h. So right after I–I lie in bed for 13 h, sometimes slept all day, like I was just paralyzed, just couldn't do a thing. in six months I'll get paralyzed 130 times and could not move.' | P13, female, 68-year-old, L3–L5 |

3.2.2. Moderate Quality of Life (n = 9)

Participants in this group described two factors that appeared to affect their quality of life negatively: feeling judged and decreasing mobility. Similar to the low quality of life group, all participants in the moderate quality of life group reported feeling judged or discriminated against. Furthermore, all nine participants stated that they have been challenged by able-bodied people and some of them reported being stared at or that others would be whispering about them.

Participant P16 described her experience with being challenged by able-bodied people in Table 4. In contrast, participant P09 described having a positive interaction with abled-bodied people (Table 4). Two participants mentioned that they understood the curiosity and the avoidance of some people towards people with a disability. Participant P20, a 60-year-old female living injured at L3–L4, indicated she is no longer surprised and does not feel it is a problem when people avoid her because she believes that abled-bodied people might not know how to approach people with disabilities.

Like those in the low-quality-of-life group, more than half of these participants reported mobility challenges. For example, P18 related how his gait deteriorated following surgery (Table 4). Participants (n = 5) in the moderate quality of life group described leg weakness. P07, a 62-year-old male who had a motor vehicle accident 45 years ago and was injured at the C6–C7 level related that before he could climb a couple of stairs but now after so many years with the spinal cord injury, he has trouble climbing stairs and would fall too many times. Sometimes he could tell if he was about to fall over, but other times his legs would just give out and he would fall over and lie flat on the ground for minutes or hours until he felt his legs again.

Participants in this group described three factors that appeared to positively affect their quality of life: learning to adapt to life after injury, appreciating their support system (family and social support), and having financial stability. Seven participants indicated how they used devices and adapted to their environments. Participant P09 explained in Table 4, how he navigated through challenges and adapted to his injury. After their injury, seven participants explained how they got used to and accepted things that they could not do. Participant 10, a 46-year-old female, injured at C5–C6 for eight years described how she learned to overcome chronic pain (Table 4). It was still difficult for four participants to adapt especially when pain is always lingering. They said that with pain, any simple task would take twice the time to complete, others related having to lie down for one hour to calm down the pain which can be related to what Participant 10 said (Table 4). Participant P10 described that her days were always planned and that there was no spontaneity in what she did because planning helped her to perform better during the day and have her routine going. She indicated that everything needed to be planned to avoid wasting energy, experiencing muscle spasms, or increasing pain.

Most of the participants in this group (n = 7) indicated family support was important for recovery. They appreciated their social support which included their family and others. P04, a 36-year-old female with a congenital spinal birth defect (C5–C6 is fused), described in Table 4 how much she felt blessed to be well supported. Four participants considered themselves lucky to have social support because this gave them the courage to keep moving in life, overcome obstacles, and maintain good mental health as expressed by participant P07 in Table 4. One participant felt fortunate to be part of a supportive community. When the wife of participant P20 (a 60-year-old male injured at L3–L4) was away at work, he reported he could rely on his neighbors for help. More than half of the participants (n = 5) did not describe having major financial barriers. Participant P10 expressed gratitude for financial stability as stated in Table 4. Five participants were grateful they were able to maintain financial stability during COVID-19 restrictions as described by Participant P09 in Table 4.

**Table 4.** Moderate life satisfaction quotes organized by factor.

| Topic | Quotation | Participant Information |
|---|---|---|
| **Feeling judged** | | |
| Feeling hyper visible | 'They don't realize maybe how disabled I am, why I'm disabled, or why I'm using a cane. So, even when I'm in a wheelchair, I always have my canes with me (move or go somewhere, where I can't go with my wheelchair), so if I stand up and use the canes, I kind of get this look, oh why are you using your canes? Why are you using a chair then? I might get a strange look.' | P16, female, 67 years old, C3–C4 |
| Expressing a positive experience with abled bodied people | 'I've been lucky because, in pickleball and ping pong and all the other sport-type activities I wanted to join, I never had to join through a disability-specific organization because I've always been accepted by able-bodied people to join as a guy in a wheelchair.' | P09, male, 67 years old, C6–C7 |
| **Experiencing a decrease in mobility** | | |
| Acknowledging a decrease in mobility after surgery | 'Before I could walk with 2 poles, but now I cannot walk with them instead I have to use a walker, without the walker it's just that I look like I'm a little drunk because my coordination is not there all the time. My equilibrium is not what it used to be, if I unexpectedly come on a change in the terrain I'm walking on, it's a big problem for me because I can't react normally and could end up falling. I look like I'm drunk really and that saddens me.' | P18, male, 54 years old, lumbar |
| **Learning to adapt to life after injury** | | |
| Navigating through challenges | 'I have turned my wheelchair into a tricycle, so Vancouver is full of hills, . . . now perfectly able to go up and down hills anywhere I want. The apartment that I have has been completely modified for accessibility, the cabinets come down to my level, I have a cooktop and a kitchen sink that can come down to my level, and the entire washroom is accessible for me in my wheelchair. So my apartment does not have barriers for me.' | P09, male, 67 years old, C6–C7 |
| Making peace with things I cannot do | 'I would be getting up most of the mornings with back pain, but I had been living with back pain the majority of the time, and I've just gotten used to it. And I functioned with it.' | P10, female, 46 years old, C5–C6 |
| Adjusting to their routine | 'My day is very limited in what I did in my day. A lot of tasks would take me twice as long to do. I have to make sure I'm not sitting too long because then the pain kicks in, but at the same time, I can't do movement for long periods. If I do go on the treadmill I need to sleep that day for 20 min to an hour.' | P10, female, 46 years old, C5–C6 |
| **Appreciating their support system** | | |
| Feeling blessed to have the support | 'I'll have meetings with my pastor who's also a good friend, so mentally, I do alright. With my boyfriend's family and abled-bodied friends I feel [I am] in an inclusive environment and it just I feel supported.' | P04, female, 36 years old, C5–C6 |
| Expressing a support system gives courage to keep moving forward in life | 'I have friends and they are always looking for places for me to rest, which is nice. We'll just sit here for a while. Because they know I have limits and I do get tired. They don't want me to fall over because then they have to pick me up—which has happened a few times.' | P07, male, 62 years old, C6–C7 |
| **Having financial stability** | | |
| Showing gratitude that partner still has a job | 'At the moment I'm very fortunate with my husband that he has a good job and he's working. At the moment we're okay financially.' | P10, female, 46 years old, C5–C6 |
| Feeling fortunate for financial stability | 'I'm very fortunate that I did not lose my job [because of the pandemic]. I worked for 5 years after my accident, then I retired. I have a good federal pension, so COVID-19 has had no impact and my accident has had minimal impact. I was well insured when I had my accident so that helped me very well. I have no financial issues that arise after my accident. I don't have to wait for approval for spending money, to go somewhere, whatever it is.' | P09, male, 67 years old, C6–C7 |

### 3.2.3. High Quality of Life (n = 10)

Participants in this group described a variety of factors that appeared to positively affect their quality of life: demonstrating resilience, learning to adapt to life after injury, appreciating their support system (family and social support, feeling included in spinal cord injury programs and community), experiencing little difference with day-to-day life during COVID-19 restrictions and having financial stability.

Many of the participants (n = 7) demonstrated resilience by emphasizing the determination they exhibited to recover after their injuries as participant P03 described in Table 5. Among the 10 participants, seven of them experienced ongoing physical improvements. P22 now can walk 50 m without any walking support (Table 5). Participant P17 a 28-year-old female who is injured at the thoracic level explained that seeing progression since her injury kept her determined to do more. She also said that not letting herself worry about the things she cannot do and just focusing more on what she can do instead also helped.

**Table 5.** High life satisfaction quotes organized by factor.

| Topic | Quotation | Participant Information |
|---|---|---|
| **Demonstrating resilience** | | |
| Expressing determination to recover from injury | 'It took me three months to make it out to the end of the driveway after I got out of the hospital. I must've fallen at least twice a day. My problem was my feet, I'd be walking along and all of a sudden one leg would go out. It took a long time to learn to keep the feet going where I wanted to, even after I managed to learn to get the feet to go where I wanted to, I had to watch it. I could go for a walk but I wouldn't see anything besides the concrete (laugh).' | P03, male, 67 years old, C5–C6 |
| Seeing progress over time | 'Recently because I'm still seeing progressing like I'm still seeing improvement in my recovery, so that's why I'm still attacking this uh as aggressively, now I can walk 50 m without needing a walking device, and I feel fortunate that I can walk.' | P22, male, 49 years old, C6 |
| **Learning to adapt to life after injury** | | |
| Facing challenges | 'I don't look at barriers, I look at them more like challenges, I usually come up with a way of um modifying something so I can still do it. I'll give you an example I started deer hunting, and my wife said, "Why don't you go and get some venison," because I grew up on that when we were young with our families. So I got my bow and go out and did that, but I couldn't carry the deer on my shoulders as I could before, so I made a kind of a sled and a harness and ski poles.' | P08, male, 72 years old, lumbar |
| **Appreciating their support system** | | |
| Being thankful for having support | 'If my wife, wasn't there, I would be struggling at this point. I really would. And so, she has been, she's been my rock type of thing. She's really helped. And I couldn't like I said I don't think I would be here if it wasn't for that because there's thoughts, you know right after, and I didn't see a future, I didn't see anything. And that was really difficult.' | P24, male, 68 years old, L3–L5 |
| Feeling a sense of belongingness to a group | 'I feel included and welcomed [at the spinal cord injury organization]. Well, I have the same problems, in terms of bowels and bladder, if you went to a family physician, they may see one or two cases in their whole time, in their practice. Whereas, the people who specialize in this, so, or the people in the ambulatory group, you're talking to people who do it 24/7, and have these problems. So, they have much better answers than, the doctors. So, it's quite good and powerful to have this cohort. Just talking to each other about the problems we face.' | P24, male, 68 years old, L3–L5 |

Similarly to the moderate quality of life group, participants (n = 8) in the high quality of life group described how they successfully adapted their lives after injury. Participant P08 related how he reinterpreted things to adapt his life according to his injury (Table 5).

Participant P02 is a 27-year-old male, who has had a C4–C6 spinal cord injury for 4 years and has a lot of determination to live his life to the fullest. He explained that if he wanted to get out into places that he used to go to, before his accident, then he would do it, and even if he must take his power-assisted wheelchair instead of the walking poles he would do it. The power-assisted wheelchair would allow him to go up and down some trails which he loved doing before his injury.

More than half of the participants in the high-quality-of-life group, like those in the moderate-quality-of-life group, valued their support system. Seven participants emphasized the importance of having family or social support to enhance mental well-being as participant P24 explained in Table 5. Six participants felt part of a community with shared beliefs, common interests, and common experiences. P24 shared that the spinal cord injury groups were very powerful (Table 5). P19, a 60-year-old male injured at the thoracic level for 13 years, related how the peer support group gave him the courage to accomplish his goals.

Half of the participants from the high-quality-of-life group experienced little difference in their daily life during COVID-19 restrictions. Participant P23 a 67-year-old male, injured at L4–L5 explained that COVID-19 had no impact on his routine because he and his wife were retired and spent most of their time at home. Two participants (P03, P21 (66-year-old female, T11–T12)) reported that they were already isolated before the COVID-19 pandemic because of their immunocompromised system. Participants (n = 5) in the high-quality-of-life group were financially stable, as were those in the moderate quality-of-life group. P12, a 69-year-old male with a C2–C3 spinal cord injury 17 years ago, expressed that he has been financially stable through the settlement money he got from his lawsuit and the money he got from a program known as a self-managed care option for home support services.

## 4. Discussion

This study explored the quality of life experiences of people with incomplete spinal cord injury who can ambulate in the context of North Americans. This is the only study that we are aware of that has identified factors associated with different levels of quality of life in this population. Also, this study is novel because we were able to outline and discover specific perceptions of participants based on their quality of life indicators. Another novelty is that it is one of the first studies in this area since the start of the pandemic. The findings emphasize the detrimental effects of COVID-19 on the quality of life of people with incomplete spinal cord injury who can ambulate, especially among those with lower levels of quality of life. We have organized our discussion as follows: contributors to low quality of life, barriers to quality of life among those with low and moderate quality of life, facilitators to quality of life among those with moderate and high quality of life, and facilitators to a high quality of life.

### 4.1. Contributors to Low Quality of Life

Our findings emphasize five main potential contributors to lower quality of life: social isolation, unemployment, secondary complications, not attending rehabilitation, and the number of years living with injury. Qualitatively, social isolation appeared to be an important contributor to low quality of life; quantitatively, living alone versus with others demonstrated a moderate effect size, which may have been exacerbated by pandemic-related lockdowns. Similar findings were reported in a cross-sectional study [13]. Potential reasons for social isolation that have been identified include challenges to participate in meaningful community activities because of problems with the built (e.g., accessibility) and social (e.g., discrimination) environment [47–49]. Low quality of life may be related to stigmatization in terms of how people with incomplete spinal cord injury are treated by others, but also in terms of how they may have internalized negative attitudes towards people with disabilities (i.e., self-stigma) [50].

Unemployment, which is a common issue among people with either complete or incomplete spinal cord injury, appeared to be associated with a lower level of quality

of life qualitatively and quantitatively (i.e., medium effect size). The finding that the majority of the participants were unemployed and depended on pensions or compensation for their day-to-day life was congruent with Canadian data that reported that 72% of people with spinal cord injury who are of working age obtain some type of financial compensation and 59% are unemployed [4]. Work-related travel has become inextricably linked to economic activity making employment and the ability to move inseparable [51]. From the results, 80% of participants identified the environment or the infrastructure as a barrier to community mobility. This corresponds with a Sweden study that reported people with spinal cord injury felt that environmental factors including infrastructure and transportation among others are key determinants of good quality of life [52]. Lack of employment may negatively affect people's sense of self-worth, may reflect problems managing secondary health conditions, and may have financial implications [53,54].

Our findings emphasize how secondary complications such as pain, neurogenic bladdebowel dysfunction, loss of sexual function, fatigue, depression, and frustration that are common in people with spinal cord injury [9–11] can have adverse effects on the quality of life. This finding is similar to the previous cross-sectional and qualitative exploration studies that reported secondary complications such as pain and spasticity may exacerbate fatigue, and in turn, have an adverse effect on emotional and physical well-being [8,22,23]. A lack of rehabilitation, which was identified as a contributor to lower quality of life in our sample, has previously been found to be associated with increased secondary complications and lower functional status, which negatively impacted the quality of life [3,55–57]. Our study findings emphasize the potential long-term benefits of rehabilitation, which demonstrated a large effect size.

There are mixed findings about the relationship between the number of years post-injury and quality of life. Some studies showed a positive relationship [58–68] and some demonstrated stability [69–73]. Others, similar to ours, show a negative relationship [69–74]. A few caveats with the previous studies are that they are quite dated (i.e., most were published over 20 years ago and they drew on longitudinal data that was sometimes collected over a 30-year time frame). Most of the participants in these studies had complete injuries, so it may be that our findings are more reflective of people with incomplete injuries. This idea was reinforced by several participants in the lower quality of life group who described perceiving they were aging prematurely.

### 4.2. Barriers to Quality of Life among Those with Low and Moderate Quality of Life

Our findings point to two other major potential barriers to quality of life: feeling judged and mobility deterioration. Participants in the low and moderate quality of life groups reported feeling judged because of challenges with their gait and ambiguity of the nature of their impairments. This negative experience is unique to this population of people with incomplete injuries who can walk compared to people with complete injuries who rely on wheelchairs. This finding is consistent with a qualitative study conducted in Australia which noted that people with incomplete spinal cord injury who can walk felt misunderstood owing to the uncertain nature of their disabilities (because wheelchairs are clear disability signifiers), reported receiving less support from the community, and described having symptoms being overlooked [11]. The prejudice they experience likely reflects the stigma associated with things like mobility assistive technology and gait abnormalities [75]. It may also be reinforced by self-stigma if people with incomplete spinal cord injury have internalized these negative public attitudes [76]. To avoid social stigma, some people with disabilities will attempt to "pass" as someone without a disability [75] which is known as passing up, and sometimes the person accentuates his disability which is known as passing down. Passing can reflect internalized oppression if it is intended to promote assimilation. It may also be an act of resistance if someone passes strategically to avoid social oppression [77].

Many participants in both the low and moderate quality of life groups experienced mobility deterioration. The ability to ambulate, which was evaluated by the functional

ambulatory category revealed a small effect size quantitatively but qualitatively mobility was an important factor. Several previous studies have reported that deterioration in the ability to walk may lead to increases in fatigue, dependence on others, and time required to complete daily activities, all of which contribute to reduced quality of life [9,11,15,66]. Previous longitudinal research has found that mobility remained unchanged initially and then declined over time among those with spinal cord injury [78]. In our study, age did not significantly differ among individuals with different levels of quality of life, however. Therefore, this does not seem to be a confounding factor with our findings. These results resonate with the findings of a prior study, which revealed that individuals who switched from walking to using wheelchairs had lower quality of life [79].

### 4.3. Facilitators of Quality of Life among Those with Moderate and High Quality of Life

Our findings emphasize three main potential facilitators of moderate and high quality of life: adapting to life after injury, social and family support, and financial stability. More than half of the participants from each group (moderate and high quality of life) adapted their lives to their injury in various ways and this, in turn, had a positive effect on their quality of life. This result is congruent with three studies on social-ecological resilience; that found that when people are faced with a lot of challenges, they have the ability to adjust to different situations in the occurrence of a change in the environment, especially unanticipated changes, in ways that continue to support the well-being of a person [80–83].

The finding that most participants in the high quality of life group were married or living with someone and that this variable had a large effect size resonates with previous studies showing similar findings [19,84–87]. These studies suggested this may be attributed to the social support couples give each other; so, the relationship might have been stronger if we had measured marital quality [88].

Peer group support and a strong community are very important for this isolated population because there is a lot of shared experience that is not seen in the general population. More than half of the participants in this group emphasized the importance of peer support, which is congruent with findings from three studies that reported in-person peer support meetings to be a highly effective way of sharing knowledge, developing skills, and decreasing secondary complications in people with spinal cord injury [87–91]. Recognizing other people with spinal cord injury as peers may promote a positive social identity. This shared identity may be a source of pride for people with disabilities [92], which may also promote quality of life.

More than 50% of participants in both groups of moderate and high quality of life described being financially stable. Although a positive relationship between income and quality of life was not found quantitatively, this may be due to the large proportion of missing data. Not surprisingly, other studies among people with spinal cord injury and other disabilities have found that not living in poverty has a positive influence on the quality of life [20,84,93].

### 4.4. Facilitators to High Quality of Life

Two possible facilitators of high quality of life are highlighted in our research: resilience and being male. Our findings suggested that resilient people led a better quality of life. Seven out of 10 participants who had a high quality of life appeared to demonstrate resilience (e.g., had a positive attitude, did not give up on challenges, and were happier). This aligns with the results of two studies that concluded resilient people are less distressed emotionally, have positive thinking, have less anxiety, and are happier [94,95]. According to research conducted on resilience, the protective mechanism that is responsible for survival is not just individual but is just as likely to be a social and ecological feature of a person's existence [96–98]. Around 70% of participants in this group adapted their routine and life to their injury which is congruent with two studies that found that people with spinal cord injury are known for their positive adjustment and tenacity, with research

suggesting that over 60% of persons with spinal cord injury adjust/adapt effectively following injury [99,100].

Being male was associated with a higher quality of life and demonstrated a large effect size. In this group, 80% of participants were male which is congruent with a study that found that being male had a positive impact on quality of life [20]. This contrasted with two previous studies that found males had poorer long-term quality of life after a spinal cord injury [66,68] which may be because people with incomplete spinal cord injury may not have the same health complications as people with complete spinal cord injury. Females with spinal cord injury experience more pain, depression [101], bowel and bladder control issues [102], urinary tract infections [102], and accessibility issues [103] compared to males. Our current finding may reflect trends in the general population, which have found that older males reported a higher quality of life than older females a finding which has been attributed to gender inequities [104].

## 5. Limitations and Future Directions

The study had three main limitations. For qualitative analysis the sample was large, but for quantitative analysis, it was relatively small and multiple comparisons were made, which increased the potential for Type I and Type II errors. The use of non-parametric tests also likely decreased the power to detect significant differences. The cross-sectional nature of the study precludes making causal attributions.

Findings from this study could inform a variety of future studies. Although we measured many relevant variables, there are other potential confounding variables, such as resiliency, which could be collected and analyzed in a study with a larger sample size. A thorough intersectional analysis could help understand how disability intersects with other identities in a way that contributes to experiences of privilege or oppression.

## 6. Conclusions

People with incomplete spinal cord injury who can ambulate share many of the same challenges as other people with complete spinal cord injuries such as secondary complications, socially isolated, and financial constraints. Those who can ambulate have some unique experiences regarding unsupportive infrastructure, unsupportive physical environment, lack of programs designed specifically for people with incomplete spinal cord injury who can ambulate, degradation in mobility, and the stigma associated with those who can potentially pass as someone without a spinal cord injury. Service providers should evaluate their programs to ensure it is inclusive of people with spinal cord injury who can ambulate and may consider offering some specific programs targeting this group [105]. For example, this could involve sensitivity training for the general population, which would help to reduce negative attitudes and misperceptions of invisible impairments and promote inclusion [106]. Findings from this study also suggest the need to improve the built and social environment for people with incomplete spinal cord injury. To reduce healthcare inequalities, changes are needed to the way the equipment and support services are provided to this population. These changes may help to address inequalities which will ultimately improve the quality of life of people with this type of spinal cord injury.

**Supplementary Materials:** The following supporting information can be downloaded at: https://www.mdpi.com/article/10.3390/disabilities3040029/s1, Table S1: Consolidated criteria for reporting qualitative research checklist 1. Table S2: Good reporting of a mixed methods study checklist 1; Document S1: Interview guide. References [27,28] are cited in the supplementary materials.

**Author Contributions:** Conceptualization, M.J., B.H., S.M., J.E., A.B., H.C., J.M., R.C., J.W. and W.B.M.; methodology, M.J. and W.B.M.; formal analysis, M.J., B.H. and S.M.; investigation, M.J., B.H. and S.M.; resources, J.M., R.C. and J.W.; data curation, M.J., B.H. and S.M.; writing—original draft preparation, M.J.; writing—review and editing, M.J., B.H., S.M., J.E., A.B., H.C., J.M., R.C., J.W. and W.B.M.; visualization, M.J., B.H., S.M. and W.B.M.; supervision, J.E., A.B., H.C., J.M. and W.B.M.; project administration, M.J. and W.B.M.; funding acquisition, M.J. and W.B.M. All authors have read and agreed to the published version of the manuscript.

**Funding:** This research was funded by the ICORD Seed Grant and the Canadian Institute of Health Research New Investigator Grant Award MSH-147809.

**Institutional Review Board Statement:** The study was conducted in accordance with the Declaration of Helsinki, and approved by the University of British Columbia's Behavioral Research Ethics Board and local health authorities (certificate no. H20-01478).

**Informed Consent Statement:** Informed consent was obtained from all subjects involved in the study.

**Data Availability Statement:** The data presented in this study are not available due to participant privacy.

**Conflicts of Interest:** The authors declare no conflict of interest. The funders had no role in the design of the study; in the collection, analyses, or interpretation of data; in the writing of the manuscript; or in the decision to publish the results.

## Appendix A

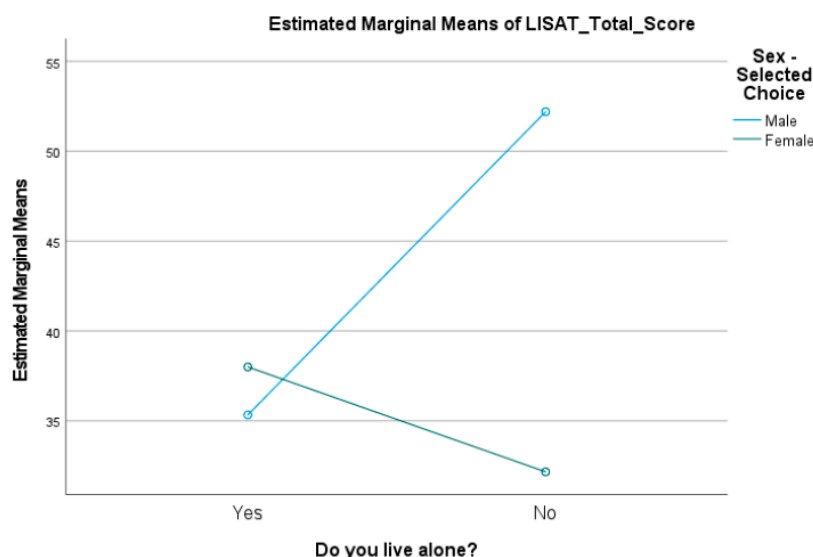

Interaction between sex and marital status

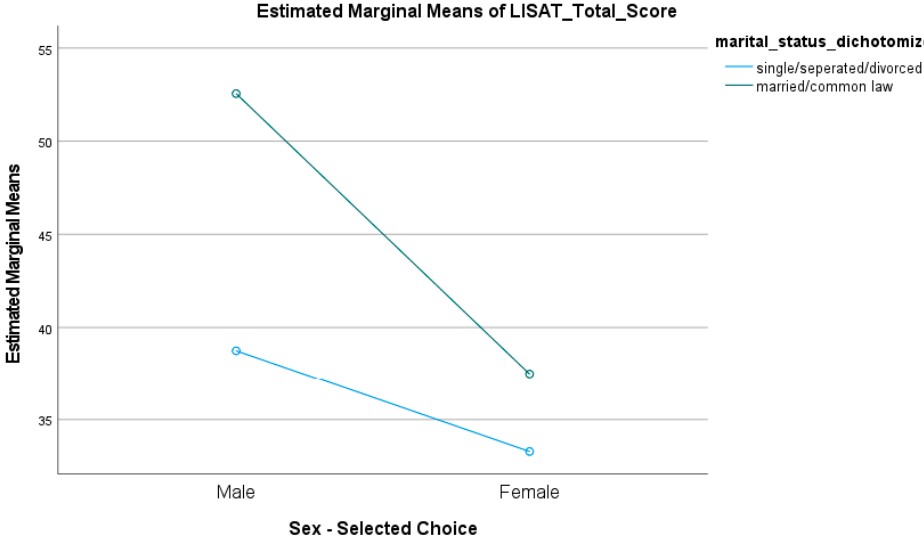

**Figure A1.** Relationship Among Sex, Marital Status, and Living Situation.

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
