# Peer review of "Exploring the Quality of Life of People with Incomplete Spinal Cord Injury Who Can Ambulate"

_disabilities, doi:10.3390/disabilities3040029_

Round 1

Reviewer 1 Report

The authors should discuss how the general population can be taught to accept, include, and facilitate people with disability to participate in main stream. Examples:

1. Education in schools and colleges to accept and respect persons with disability instead of looking down upon them.

2. Education of public via television programmes to understand the feelings of people with disability

3. Legislation to enforce employers both in private and public sector to provide opportunities for disabled persons to gain training and take up jobs. To compel employers via legislation to provide additional facilities for disabled persons to work e.g. providing suitable toilet facility, access, remote working, etc. 

4. To compel all businesses to provide access to persons with disability.

Author Response

Please see attachment :)

Reviewer 2 Report

In this manuscript, the authors focus on a survey of quality of life of incomplete spinal cord injuries with ambulation and their potential associated factors. They did quantitative and qualitative analysis and identified both negative barriers and facilitators. Finally, they propose how to improve quality of life.

It is a comprehensive study and the data can be used by medical staff, communities, people surrounding to disabled people and disabled people themselves. 

There are a few points below can be considerate to improve  quality of this study

Method:

136: ambulatory dependent on physical assistance (level 1), ambulator dependent on physical assistance (level 2),

What is difference between Level 1 and 2?

Results: The low employment (16.7%) is a big concern, which is closely related to QOL

Need to find the potential reasons why the majority of participants in the low quality of life group are female? Does it relate to neurogenic bladder and bowel that need additional attention comparing to male.

Discussion

457, “more than half of the participants were unemployed” The table 1 show 83% un-employed, and this fact should be emphasized as “the majority (83.3%) of the participants were unemployed.”

469, secondary complications should include neurogenic bladder and bowel, as well as loss of sexual function, which are very serious of complications. The participates may not disclose these complications due to sensitivity of these topics.

 508, mobility deterioration might relate to aging. 
